# First insights into deep convection by the Doppler velocity measurements of the EarthCARE Cloud Profiling Radar

Aida Galfione<sup>1</sup>, Alessandro Battaglia<sup>1,2</sup>, Bernat Puigdomènech Treserras<sup>3</sup>, and Pavlos Kollias<sup>3,4</sup>

Correspondence: Aida Galfione (aida.galfione@polito.it)

#### Abstract.

5

Convective updrafts and downdrafts play a vital role in Earth's energy and water cycles by modulating vertical energy and moisture transport and shaping precipitation patterns. Despite their importance, the characteristics of convective motions and their relationship to the near-storm environment remain poorly constrained by observations.

Doppler radars, in principle, are able to measure the vertical air motion within clouds, thus providing critical insight into convective dynamics and enabling estimates of convective mass flux. The payload of the recently launched EarthCARE satellite mission includes a 94-GHz Cloud Profiling Radar (CPR) with Doppler capability. In this study, we present first-light CPR Doppler velocity observations in deep convective clouds. These early examples offer a first glimpse into the dynamic nature of cloud systems. The narrow footprint of the CPR helps reduce the impact of multiple scattering and non-uniform beam filling (NUBF) on the Doppler velocity measurements. However, the instrument's low Nyquist velocity presents a significant challenge for recovering the true Doppler velocity profiles in deep convective systems.

The CPR Doppler velocity observations are expected to challenge traditional methodologies for identifying deep convective cores, which typically rely on reflectivity-based thresholds. We showcase examples that demonstrate the synergy between CPR Doppler velocity measurements and geostationary satellite observations, illustrating how their combined use can help capture the evolution of the convective lifecycle.

These results align with EarthCARE's broader mission objectives and highlight the potential of spaceborne Doppler radars to significantly advance our understanding of cloud dynamics and convection in the climate system.

#### 1 Introduction

Deep convective clouds are responsible for the vertical transport of air and water, one of the most influential atmospheric processes that is yet poorly constrained by measurements. Deep convection is crucial in balancing the Earth's heat budget and influencing large-scale weather patterns, including cloud formation and the development of storms and extreme weather (Hartmann et al. 1984). Deep convective events typically occur in tropical regions, but they affect the global atmospheric circulation beyond the tropics via anvil detrainment processes and latent heat release via precipitation (Hartmann et al. 2018;

<sup>&</sup>lt;sup>1</sup>Department of Environment, Land and Infrastructure Engineering, Politecnico di Torino, 10129 Turin, Italy

<sup>&</sup>lt;sup>2</sup>Earth Observation Science Group, Department of Physics and Astronomy, University of Leicester, Leicester LE1 7RH, UK

<sup>&</sup>lt;sup>3</sup>Department of Atmospheric and Oceanic Sciences, McGill University, Montreal, H3A 0B9, QC Canada

<sup>&</sup>lt;sup>4</sup>School of Marine and Atmospheric Science, Stony Brook University, NY 11790, NY USA

Gasparini et al. 2021). A number of microphysical processes are active during convective initiation and development that are not yet well understood or properly implemented in models (Prein et al. 2015; Arakawa 2004; Bony et al. 2015).

Despite the importance of deep convection, several aspects of deep convective clouds remain challenging to represent in high-resolution models (Fridlind et al. 2017; Ladino et al. 2017). Models also struggle to accurately represent convective updrafts, leaving significant observational gaps (Varble et al. 2014). Surface and airborne radar observations have provided valuable insight into the structure and magnitude of convective updrafts, but the observational record is very sparse and mostly available over land (Giangrande et al. 2013; J. Yang et al. 2016; North et al. 2017; Oue et al. 2019; Wang et al. 2020; Jeyaratnam et al. 2021). On the other hand, satellite observations can provide global coverage and sufficient sampling of deep convection and associated clouds and precipitation (Lee et al. 2021). In particular, the 3-D structure of deep convective clouds has been extensively studied using observations from spaceborne radars.

The Tropical Rainfall Measuring Mission (TRMM), developed by the National Aeronautics and Space Administration (NASA) and the National Space Development Agency of Japan (NASDA), introduced the first spaceborne radar, a 13.8 GHz Precipitation Radar (PR) (Kummerow et al. 1998; Kummerow et al. 2000). The TRMM PR was operational from 1997 to 2015 and advanced our understanding of tropical convection and associated rainfall (Xu et al. 2012; Yokoyama et al. 2014). Studies using the TRMM PR data analyzed the structure of convective systems, including diurnal cycles and vertical profiles (Hamada et al. 2015). TRMM's success led to the Global Precipitation Measurement (GPM) mission launched in 2014 by NASA and the Japan Aerospace Exploration Agency (JAXA) which enhances TRMM's capabilities with improved resolution and higher latitude coverage (Skofronick-Jackson et al. 2017). The GPM mission features a Dual-frequency Precipitation Radar (DPR) that operates at Ka (35.5 GHz) and Ku (13.6 GHz) bands, providing multi-frequency measurements of 3D precipitation structures (Skofronick-Jackson et al. 2018). Studies using GPM DPR data show deep convection reaching the tropopause predominantly over land, consistent with TRMM findings (Liu et al. 2016; Liu et al. 2020; Battaglia et al. 2020). Ni et al. 2019 analyzed ice microphysical properties, revealing larger ice particles and higher ice water content in land-based deep convective cores. The limited sensitivity of the PR and DPR limited their ability to capture the 3D structure of the upper-level tropospheric cloud structures.

The CloudSat-CALIPSO mission (Stephens et al. 2002), part of NASA's A-Train since 2004, provided detailed cloud vertical structures. Its Cloud Profiling Radar (CPR) with 240 m vertical resolution captured convective cloud features, aiding studies on convective cores and updrafts (Takahashi et al. 2017). Findings indicate stronger convective cores and lower entrainment rates over land, enabling higher-altitude particle transport. Because CloudSat was a nadir-looking radar, not scanning across its track, it was limited in its ability to capture 3-dimensional spatial heterogeneity of deep convective cores (DCCs). To mitigate biases, CloudSat data have been integrated with passive sensors, such as MODIS cloud top temperature, for improved convective characterization (Luo et al. 2008; Luo et al. 2010; Luo et al. 2014; K. Yang et al. 2023).

50

55

Launched in 2018 as a NASA mission, RainCube demonstrated the feasibility of operating a scientific Ka-band radar from a CubeSat platform, with the radar instrument developed at NASA's Jet Propulsion Laboratory (JPL) (Peral et al. 2018). Its innovative deployable antenna enabled compact integration and lower launch costs, paving the way for constellations of radar-equipped satellites, as the INCUS train formation.

Finally, in May 2024, the Earth, Cloud, Aerosol and Radiation Explorer (EarthCARE, Illingworth et al. 2015), a joint European Space Agency (ESA) and JAXA mission, was successfully launched. The EarthCARE (EC) mission aims to improve cloud-aerosol-radiation interaction studies and enhance numerical weather prediction (NWP) models and climate simulations. EarthCARE carries a 94-GHz Doppler Cloud Profiling Radar (EC-CPR), a High-Spectral Resolution Lidar (ATLID), a Multi-Spectral Imager (MSI), and a Broad-Band Radiometer (BBR). Launched after CloudSat-CALIPSO ended operations in 2023, EarthCARE benefits from an improved radar sensitivity owing to its lower orbit and from having all instruments on the same platform (Illingworth et al. 2015; Wehr et al. 2023). Most importantly, the EarthCARE mission features the first spaceborne radar with Doppler capability (Amayenc et al. 1993; Kobayashi et al. 2002; Meneghini et al. 2003; Kollias et al. 2014; Kollias et al. 2018; Kollias et al. 2022a). The availability of Doppler measurements from space offers a unique opportunity for the collection of a global dataset of vertical motions in clouds and precipitation. This global data set is expected to improve our understanding of convective motions in clouds and help evaluate current parameterizations of convective mass flux in cloud resolution models (Manabe et al. 1964; Tiedtke 1989; Bechtold et al. 2001).

Here, a first assessment of the performance of the EC-CPR Doppler velocity measurements in deep convection is presented. The main objectives of this study are to describe and interpret convective cores as observed by the EC-CPR, leveraging joint Doppler velocity and reflectivity measurements, and to compare these observations with geostationary data. For the first time, Doppler velocities from a spaceborne radar are used to identify and characterize convective cores, providing insights into their internal dynamics and updraft structures (Kollias et al. 2023). Coincident MSI observations are compared with geostationary Meteosat Second Generation (MSG) imagery to assess the capability of passive sensors to detect convection and track its evolution.

#### 2 CPR Doppler velocity observations in deep convection

One of the new capabilities of the EarthCARE mission is the CPR Doppler velocity measurements. Several factors are expected to impact the quality of the CPR Doppler velocity measurements (Tanelli et al. 2002; Tanelli et al. 2005; Schutgens 2008a; Schutgens 2008b; Kollias et al. 2014; Kollias et al. 2018; Hagihara et al. 2022; Kollias et al. 2022b). The EarthCARE satellite speed of 7.6 km/s introduces significant broadening (decorrelation) of the CPR phase measurements that causes significant uncertainty in the Doppler velocity estimates (Kollias et al. 2014; Kollias et al. 2022b). Antenna mispointing is another source of uncertainty (Tanelli et al. 2005; Battaglia et al. 2014; Puigdomènech Treserras et al. 2025). In deep convection, additional factors such as attenuation, multiple scattering (Battaglia et al. 2008; Battaglia et al. 2010; Battaglia et al. 2011c), non-uniform beam filling (Tanelli et al. 2002; Kollias et al. 2022b), and aliasing (Sy et al. 2014) can have a significant impact on the observed Doppler velocities and introduce considerable uncertainty and biases.

#### 2.1 Convection embedded in stratiform event

85

An example of CPR observations of a deep convective system is shown in Fig. 1. The CPR observations were collected on September 19, 2024, over Western Africa on a descending (daytime) orbit. Here, CPR Level 2a (L2a) C-PRO data products

**Figure 1.** (a) CPR reflectivity during a large-scale, deep precipitating system with embedded convection observed on September 19, 2024 over the Tropical Western Pacific (Frame 1760E). The horizontal line indicates the 10 km height, and the blue circles indicate the maximum height where a dBZ value above 10 dBZ is observed. (b) The CPR Doppler velocity measurements after a 4-km along-track integration (Kollias et al. 2023). Positive Doppler velocities indicate hydrometors' movement towards the ground.

are used (Kollias et al. 2023). These products are derived from the CPR Level 1b data plus auxiliary meteorological data. The L2a C-PRO data product was released to the research community in March 2025 (Eisinger et al. 2023). The CPR reflectivity factor (Fig. 1a) illustrates the vertical structure of a wide deep precipitating system. The reflectivity plot in Fig. 1a has 1 km resolution in the along-track dimension, 100 m resolution in the vertical dimension. CloudSat-based studies of deep convection mainly use the reflectivity profile features near cloud top to identify deep convective cores (DCC, Takahashi et al. 2012; Luo et al. 2014; Takahashi et al. 2017; Stephens et al. 2024). The underlying reasoning is that the overshooting of radar reflectivity is an indicator of the larger-size particles pushed higher up; this is only possible with the presence of strong rising updrafts. Three criteria are commonly adopted for convection identification (Takahashi et al. 2014):

- CPR cloud mask (2B-GEOPROF product) greater than 20;

100

- A continuous radar echo from below 2 to above 10 km, thus a thick cloud deck;
- The 10 dBZ echo top height which is indicative of the level where large size particles are lofted by strong convection above 10 km (Luo et al. 2008).

In Fig. 1a, the 10 dBZ echo top height is very close to the 10 km height for a significant part of the deep precipitating system. In two areas (1710-1730 km and 1890-1910 km along track), the 10 dBZ echo top height is well above the 10 km height and

closer to the cloud top height. Luo et al. 2014 introduced a fourth criterion for detecting DCCs, which requires that the 10 dBZ echo top height be within 2 km of the cloud top height determined by the CPR.

110

125

130

The CPR Doppler velocity measurements for the same event can help us evaluate these different methodologies for identifying DCCs. Figure 1b shows the CPR Doppler velocity averaged over a 4-km along-track distance. The CPR Doppler velocity measurements are shown only in areas where the CPR reflectivity exceeds -15 dBZ. The native CPR along track resolution is 500 m, thus, a total of nine CPR Doppler velocity estimates (their respective real and imaginary parts of the lag-1 pulse pair estimator) have been averaged (Kollias et al. 2023). The averaging operation has to be performed in the lag-1 space, in order to avoid the cancellation due to opposite sign in the velocity space, that would lead to a wrong estimation of the Doppler velocity. Averaging over a larger number of pulses reduces aliasing but does not eliminate it, meaning that the Doppler velocity estimates remain susceptible to aliasing errors. Conversely, using a 4-km integration length constrains the ability to resolve the variability within convective cores, which typically occurs at sub-kilometer scales. Before the along track averaging, the CPR Doppler velocities have been corrected for antenna mispointing (Puigdomènech Treserras et al. 2025) and non-uniform beam filling (NUBF) Doppler velocity biases (Kollias et al. 2014; Sy et al. 2014).

The nadir-pointing CPR Doppler velocity  $V_D$  represents the sum of the vertical air motion  $W_{AIR}$  and the reflectivity-weighted Doppler sedimentation velocity of the hydrometeors  $V_T^D$ :

120 
$$V_D = W_{AIR} + V_T^D$$
. (1)

The term  $V_T^D$  can only take positive values (downward motion) while the term  $W_{AIR}$  can take both positive (downdraft) and negative (updraft) values. The majority of the observed  $V_D$  in Fig. 1b are positive. This implies that the magnitude of  $V_T^D$  is higher than that of the embedded  $W_{AIR}$  updrafts. This suggests the presence of negligible vertical air motions ( $|W_{AIR}| 

**Figure 2.** (a) The CPR Doppler velocity profiles at along track distance of 1840 km. The 4-km CPR Doppler velocity estimate is shown as green circles and the 1-km Doppler velocity estimates within a 2 km distance from 1840 km are shown as gray lines. The yellow vertical lines indicate the CPR Nyquist velocity and the horizontal dashed line indicates the melting layer height. (b) The corresponding CPR reflectivity at along track distance of 1840 km.

Doppler velocity estimates (Fig. 2a). Below the melting layer, velocity folding occurs especially in the 1-km CPR Doppler velocity estimates, which are noisier. In Fig. 2a the 1-km Doppler velocity estimates outside the  $V_N$  boundaries have been corrected for velocity folding. The assumption used for the unfolding is that negative Doppler velocities below the melting layer in a stratiform precipitation profile are the results of  $V_D$  exceeding  $+V_N$ . Subsequently, all negative  $V_D$  values below the  $0^{\circ}$ C isotherm are unfolded by adding  $2 V_N$  to them.

The interpretation of the CPR Doppler velocity profile in deep stratiform layers serves as a baseline for understanding convective updrafts. In Fig. 1b, updrafts are depicted as regions with negative (upward) 4-km-averaged  $V_D$  estimates in cold temperatures (Fig. 1b). The clusters of negative  $V_D$  are generally located near the cloud top, with the exception of the cluster located at along-track distances between 1890 and 1910 km. Since ice particles are smaller at colder temperatures, it is plausible that near cold cloud tops, weak gravity waves and updrafts contribute to an overall negative (upward) CPR Doppler velocity signal. Interestingly, two regions around 1720 and 1900 km with 10 dBZ echo top height well above the 10 km altitude exhibit such dynamical features. At 1890-1910 km along-track, a deep and coherent dynamical structure is observed, characterized by strong upward motions extending from 8 to 14 km. This vertically oriented feature represents a deep convective updraft and is collocated with the highest 10 dBZ echo top heights. The  $W_{AIR}$  within this convective updraft is strong enough to cause velocity folding, depicted as a red patch of Doppler velocities embedded within the negative Doppler velocity cluster.

**Figure 3.** (a) CPR reflectivity during a deep convective event on September 18, 2024 over Western Africa (Frame 1752E). The blue circles indicate the height where multiple scattering effects become important. The vertical dashed lines indicate the locations where CPR profiles will be shown in later figures. (b) The CPR Doppler velocity measurements after a 4-km along-track integration (Kollias et al. 2023). Positive Doppler velocities indicate hydrometors' movement towards the ground. The black contour indicates the area where the 4-km CPR Doppler velocity standard deviation exceeds 2 m/s. A box of 3 km along-track by 2 km in range is used for the estimation of the standard deviation.

#### 2.2 Deep convective scene

155

160

165

The complexity of the  $V_D$  profiles in deep convection is examined using a sample deep convective cloud (DCC) observed by the CPR (Fig. 3). The DCC is located between 1265 and 1300 km along track and is characterized by overshooting cloud tops reaching up to 17 km in altitude. Strong attenuation is observed (Fig. 3a), and the smooth appearance of radar reflectivity echoes extending to and below the surface indicates the presence of moderate multiple scattering effects (Battaglia et al. 2010). Regions contaminated by multiple scattering are currently identified in the C-FMR product (Kollias et al. 2023) using a simple flagging approach based on the methodology proposed by Battaglia et al. 2011a. The blue-filled circles indicate the height at which multiple scattering effects on radar reflectivity are expected to become significant. To correctly interpret Doppler velocities in deep convection, it is essential to assess the influence of multiple scattering on the Doppler signal (Battaglia et al. 2011b). Doppler velocity measurements within regions affected by multiple scattering cannot be considered reliable. Although a marked reduction in the correlation between successive pulses—and consequently an increase in Doppler velocity noise—is expected in these regions, such behavior is not always observed. This inconsistency warrants further investigation in future studies. Adopting a conservative approach, Doppler velocity values within these portions of convective cell profiles should therefore be treated with caution or excluded from quantitative analysis. However, since this issue is beyond the scope of the

Figure 4. (a) The CPR reflectivity profile at along-track distance of 1268 km. The yellow filled circles section of the CPR reflectivity profile indicate the CPR gates where the Doppler velocity estimates are considered unaffected by multiple scattering. The green triangle indicates the height of the maximum radar reflectivity. (b) The 4-km CPR Doppler velocity profile (blue filled circles) and the 1-km CPR Doppler velocity profile (orange crosses). The black dashed vertical lines indicate the CPR Nyquist Doppler velocity. (c) The unfolded 4-km CPR Doppler velocity profile (blue filled circles) and the unfolded 1-km CPR Doppler velocity profile (orange crosses). The black dashed vertical lines indicate the CPR Nyquist Doppler velocity. The red shading highlights the sections of the profile affected by multiple scattering.

present study, our interpretation is limited to the portion of the  $V_D$  profiles above the height at which multiple scattering effects are expected to become significant.

Figure 3b shows that the  $V_D$  profiles vary substantially with regions of both positive and negative values. In this frame the Nyquist velocity is 5.08 m/s and the PRF is  $6.38\,\mathrm{kHz}$  The range of  $V_D$  values spans the full Nyquist interval  $[-V_N:+V_N]$ . The convective  $V_D$  profiles are characterized by frequent Doppler velocity aliasing. Fig. 3b presents the 4-km averaged  $V_D$ . Velocity aliasing is even more pronounced at the 1-km averaged  $V_D$ . The observed variability of  $V_D$  serves as a strong indicator of the presence of convective updrafts and downdrafts. In Figure 3b, the black outline highlights regions where the standard deviation of Doppler velocity exceeds 2 m/s. The standard deviation is calculated within a moving window of 3 km horizontally and 2 km vertically, centered on each pixel, to capture Doppler velocity variations in both the along-track and across-track Doppler velocity directions.

Two example profiles corresponding to the along-track locations indicated by the black dashed lines in Fig. 3b are analyzed to explore the complexity of  $V_D$  in deep convective cores. The first profile is shown in Fig. 4. The CPR reflectivity profile is presented in Fig. 4a. The yellow-filled circles mark the CPR range gates where Doppler velocity estimates are considered unaffected by multiple scattering. Additionally,  $V_D$  estimates near the cloud top are excluded if the reflectivity falls below -15 dBZ. The maximum reflectivity is observed at an altitude of 11 km, more than 5 km below the cloud top. The corresponding  $V_D$ 

Figure 5. (a) The CPR reflectivity profile at along track distance of 1283 km. The yellow filled circles section of the CPR reflectivity profile indicate the CPR range gates where the Doppler velocity estimates are considered unaffected by multiple scattering. The green triangle indicates the height of the maximum radar reflectivity. (b) The 4-km CPR Doppler velocity profile (blue filled circles) and the 1-km CPR Doppler velocity profile (orange crosses). The black dashed vertical lines indicate the CPR Nyquist Doppler velocity. (c) The unfolded 4-km CPR Doppler velocity profile (blue filled circles) and the unfolded 1-km CPR Doppler velocity profile (orange crosses). The black dashed vertical lines indicate the CPR Nyquist Doppler velocity. The red shading highlights the sections of the profile affected by multiple scattering.

profiles, averaged over 1-km and 4-km along-track intervals, are shown in Fig. 4b. The black dashed lines indicate the Nyquist bounds, while the vertical yellow line indicates zero Doppler velocity. As expected, the 4-km-averaged  $V_D$  varies less with height compared to the 1-km  $V_D$  estimates. This vertical correlation is expected, given that the CPR pulse length is 500 m and  $V_D$  is estimated every 100 m.

Here, we focus on interpreting the  $V_D$  estimates within the section identified as having reliable Doppler velocity observations. Beginning with the 4-km profile: near the cloud top, the value of  $V_D$  is negative, indicating the presence of a weak updraft. Below 14 km, the value of  $V_D$  turns positive, which may indicate the presence of large hydrometeors with high sedimentation velocity and/or a downdraft, resulting in an apparent downward motion. The abrupt jump of about 10 m/s in the profile at 12.5 km is attributed to velocity aliasing. In general, if the absolute value of the difference between two consecutive Doppler measurements exceeds the Nyquist velocity, then adding  $\pm 2 V_N$  to one of the velocity produces a smoother profile. Due to the noisiness of the measurements, the identification of a fold is not straightforward and there will be some ambiguity for successive points in the profile with jumps in  $V_D$  close to  $V_N$  (e.g. at 4-km integration length, values within 1 m/s from  $V_N$  are potential foldings). In this example, the difference is much larger, so folding is identified unambiguously and unfolding is straightforward. The entire segment of the profile between 9 and 12.5 km is therefore aliased; Fig. 4c shows the unfolded 1-km and 4-km  $V_D$  profiles. The aliased section of the 4-km profile has been corrected by adding 2  $V_N$ . The unfolded 4-km profile

190

**Figure 6.** (a) The normalized frequency of occurrence of CPR Doppler velocity folding, (b) The NUBF induced CPR Doppler velocity bias in m/s in convective and stratiform regions, derived by multiplying the along track reflectivity gradient by a correction coefficient (0.165  $m \cdot km/s \cdot dB$ ) (Kollias et al. 2023).

displays a smooth vertical structure. Except for a small region near the cloud top, the  $V_D$  values remain positive, suggesting that in this upper part of the convective tower all hydrometeors are falling to the ground.

The second profile is shown in Fig. 5. The CPR reflectivity profile is presented in Fig. 5a. This profile is selected from the elevated cloud top region of the deep convective cloud to highlight the complexity of the Doppler velocity measurements, especially in convective regions where multiple scattering effects are significant. In this case, the maximum CPR reflectivity is detected higher in the profile, only 2.5 km below the cloud top. The corresponding  $V_D$  profiles, averaged over 1-km and 4-km along-track intervals, are shown in Fig. 5b. These  $V_D$  profiles appear more complex due to the presence of significant multiple scattering effects. Fig. 5c shows the unfolded 1-km and 4-km  $V_D$  profiles. In this case, the section of the 4-km averaged  $V_D$  from the cloud top to a height of 13.8 km is identified as aliased and corrected by subtracting 2  $V_N$ . The unfolded 4-km profile displays a smooth vertical structure. A strong updraft is present above 12 km, and its magnitude exceeds 10 m/s near the cloud top.

## 2.3 Analysis of Doppler velocity aliasing

The analysis of the two convective  $V_D$  profiles underscores the challenges associated with unfolding CPR Doppler velocity profiles in deep convection. The low Nyquist velocity of the EarthCARE CPR ( $V_N 

Figure 7. (a) Radiance from channel 1 (0.6 µm) of MSG, on November 7th, 2024 at 13:45 UTC. The EarthCARE ground track, corrected for parallax is shown as the red line. (b) The MSI IR channel data from the EarthCARE satellite. The overpass time is 13:43 UTC on November 7th, 2024, frame 2530D. The red line is the satellite ground track. The segment between the two stars is plotted in Fig. 8.

foldings per profile, while the blue line indicates the number of pixels per profile that exceed the stratiform range threshold ([-2 3] m/s).

285

290

295

Using the CloudSat methodology, a DCC would have been identified in Cell 1 between 38.5° and 38.7° of latitude (green bar on the right in Fig. 8a) whereas only the central tower in Cell 2, located around 36.5° of latitude, would be classified as deep convective (green bar on the left of Fig. 8a). The other convective cores do not meet the required criteria for cloud-top echo height and echo continuity.

To place the CPR convective cloud snapshots within the context of their lifecycle, observations from EUMETSAT's MSG satellite are analyzed. Figure 9 and 10 display the corresponding MSG SEVIRI 1.5 rapid scan frames from channel 9 (10.8 µm) captured at 5-minute intervals before and after the EarthCARE overpass for the two convective cells analyzed in this study. In these figures, the solid black line represents the EarthCARE ground track, corrected for parallax using cloud-top height derived from radar data, while the dashed line shows the original, uncorrected ground track position. The markers correspond to feature locations shown in Fig. 8a, with the black star indicating the position of the minimum brightness temperature tracked within the cell. Strong updrafts, including overshooting tops (Khlopenkov et al. 2021), are expected to be well captured by geostationary sensors. However, this assumption may not hold in cases where convection is embedded within a thick cloud deck or occurs beneath an extensive anvil cloud, where updrafts are obscured and not directly observable by spaceborne infrared and visible passive instruments.

**Figure 8.** EarthCARE overpass 2530D over South Mediterranean sea and Atlas mountains on November 7th, 2024, at 13:43 UTC. (a) Reflectivity in dBZ, clutter removed. The green bars correspond to profiles labelled as convective from CloudSat methodology described in Takahashi et al. 2014. (b) Doppler velocity, corrected for antenna pointing and NUBF. Black contour is the standard deviation (calculated in a window 3 km horizontally and 1.1 km vertically) that exceeds 2 m/s. (c) Number of foldings per profile and number of pixels per profile that exceed the stratiform interval [-2 3] m/s.

The Tracking and Object-Based Analysis of Clouds *tobac* algorithm is a robust and well-supported algorithm for feature detection and tracking of convective clouds (Heikenfeld et al. 2019; Sokolowsky et al. 2024). In this study, *tobac* is applied to

**Figure 9.** Successive images depicting the time evolution of the IR brightness temperature field 10 minutes before (a), 5 minutes before (b), closest in time (c), 5 minutes after (d) and 10 after (e) EarthCARE overpass 2530D on November 7th, 2024, zoom on cell 1. The colors represent the brightness temperature from channel 9 (10.8 µm), measured by MSG rapid scans. Black solid line represents the ground track of EarthCARE, corrected for parallax (dashed line is the original ground track). The black markers correspond to Fig. 8a. The black star is the minimum of the brightness temperature that is tracked.

MSG imagery to track the minimum brightness temperature within Cells 1 and 2 (Fig. 11). As shown in Fig. 9, Cell 1 is present well before the EarthCARE overpass and is already in a mature stage of development. Although the cloud top is very cold, no significant cooling is detectable in association with the ongoing embedded convective updraft observed by the CPR. It is likely that, as the cell began detraining mass into the anvil, multiple sparse convective cores developed beneath it. In such cases, radar observations are essential for accurately identifying and characterizing convection. According to the geostationary tracking, at the time of the EarthCARE overpass (13:43 UTC), the minimum brightness temperature in Cell 1 is already below 215 K (Fig. 11a) and fluctuates only slightly—by a few kelvin—during the minutes surrounding the overpass. At this stage of the convective lifecycle, the evolution of the minimum brightness temperature is no longer representative of fine-scale structures such as overshooting tops or highly localized, intense updrafts—features that, in contrast, are well captured by the EC-CPR.

305

315

In contrast, Cell 2 is more isolated, which facilitates more effective tracking and allows for its evolution to be observed from the early stages. Figure 10 reveals a secondary cold spot on the southwest side of the cell, which later merges with the main convective core. The *tobac* tracking algorithm identifies a single feature, prioritizing the cooling associated with the main core while disregarding the cloud-top cooling of the smaller, secondary feature.

A significant cooling phase is observed during the first 20 minutes of the cell's development. Following this initial phase, the *tobac*-tracked cloud-top temperature remains nearly constant, plateauing at approximately 220 K over the subsequent hour.

Figure 10. Successive images depicting the time evolution 10 minutes before (a), 5 minutes before (b), closest in time (c), 5 minutes after (d) and 10 after (e) EarthCARE overpass 2530D on November 7th, 2024, zoom on cell 2. The colors represent the brightness temperature from channel 9 (10.8 µm), measured by MSG rapid scans. Black solid line represents the ground track of EarthCARE, corrected for parallax (dashed line is the original ground track). Red markers correspond in shape to Fig. 8a. The black star is the position of the minimum brightness temperature that is tracked.

The EarthCARE overpass (indicated by the red dashed line in the time series in Fig. 11a) occurs when the cell is already in its mature phase. Once again, it is challenging to directly correlate the CPR data—offering detailed vertical cross-sections of internal cloud structure—with the geostationary observations, which characterize the average behavior of the convective system based on cloud-top cooling rates. While *tobac* tracking provides valuable temporal context, it cannot capture the fine-scale vertical variability and internal dynamics revealed by the EC-CPR.

This discussion reinforces the limitations of relying solely on geostationary infrared cooling rates to characterize convection. While IR observations are effective at capturing relatively large and isolated updrafts near cloud tops, embedded convection and sub-kilometer-scale vertical motions largely go undetected. Resolving these features requires spaceborne radar observations, such as those provided by the EarthCARE CPR.

# 4 Conclusions

320

Spaceborne radar observations—such as those collected during NASA's CloudSat and RainCube missions and the NASA-JAXA TRMM and GPM missions—have provided valuable global observations of storm and convective cloud reflectivity structures. However, direct observations of convective dynamics at the global scale have been lacking until now. The recently

**Figure 11.** Minimum brightness temperature (in K) within the cell, as detected and tracked with tobac. Red dashed line corresponds to the EC overpass time. (a) Cell 1. (b) Cell 2.

launched ESA-JAXA EarthCARE mission, equipped with a Doppler-capable radar, fills this critical observational gap and marks the beginning of a new era of satellite-based radar measurements to improve our understanding of convective dynamics.

Before launch, there were numerous questions regarding the quality of Doppler velocity measurements in deep convection, particularly due to anticipated challenges such as attenuation, multiple scattering, non-uniform beam filling (NUBF) effects, a narrow Nyquist velocity range, and the vertical and horizontal variability of convective cores. In this study, EC-CPR transects across various convective systems have been analyzed to assess and illustrate the impact of these challenges on the interpretation of Doppler velocity profiles.

The availability of Doppler velocity measurements from space provides valuable new insights into the presence, as well as the horizontal and vertical extent, of convective updrafts and downdrafts. Our case studies give evidence of differences in the detection of convective cores when Doppler velocity based instead of reflectivity based criteria are used. Future studies should investigate the impact of new Doppler velocity based criteria onto the climatology of occurrences of convective cores across Earth.

Furthermore, when combined with co-located infrared observations from geostationary satellites, CPR Doppler measurements offer new perspectives on the use of cloud-top cooling rates—computed as time derivatives of brightness temperature—as proxies for convective intensity.

Some final conclusions of this work are summarized in the following.

1. The first images of Doppler velocities measured by the EarthCARE Cloud Profiling Radar (EC-CPR) offer an unprecedented view of convective motions on a global perspective. While these images reveal the presence of convection, the quantitative interpretation of the CPR signal—such as the estimation of updraft and downdraft velocities or convective mass fluxes—will require further analysis. This need arises from the inherent complexity of convective dynamics, compounded by signal noise and the limitations imposed by the narrow Nyquist velocity range.

375

- The CPR Doppler velocity measurements will serve as the foundation for a dynamics-based convection identification algorithm, designed to augment existing reflectivity-based detection methods. As demonstrated in the case study, parameters such as the standard deviation of Doppler velocity computed within a 3 km horizontal by 2 km vertical window, or the frequency of Nyquist velocity foldings, can serve as reliable indicators of convective activity.
- 2. The development of a robust algorithm for unfolding CPR Doppler velocity (VD) measurements in deep convective clouds is currently underway. The first step is to characterize the complexity of the VD field and to identify the primary sources of velocity discontinuities in deep convection. Initially, the focus will be limited to convective profiles exhibiting fewer than three Doppler velocity foldings at the 4-km along-track resolution—an approach expected to encompass more than 99% of the observed CPR VD profiles. In cases where velocity aliasing is not observed in the 4-km averaged VD, but is present in the 1-km averaged profile, the 4-km averaged VD can be used as a weak constraint to unfold the 1-km averaged VD by minimizing the difference between the two. In more complex cases, such as those shown in this study, the morphology of the CPR reflectivity profile will be used to determine the vertical continuity of the convective column. In addition, VD estimates at 500 m (native CPR along track resolution), 1-km or 4-km will be combined for the estimation of the unfolded CPR VD profile.
- 3. The CPR provides a unique capability for observing embedded convection and sub-kilometer-scale convective cells, thereby overcoming key limitations of convective observations derived from geostationary imagery. In particular, convective motion estimates based on cloud-top cooling rates are effective primarily for updrafts that are both comparable in size to the geostationary sensor's resolution (typically larger than 2 km at mid-latitudes) and located near the cloud top. As such, this method is generally limited to convective cells in the early stages of development or to those exhibiting overshooting tops.
  - 4. Geostationary imagery, on the other hand, offers significant potential for providing the spatio-temporal context of convection—such as whether it is part of a mesoscale system or an isolated cell, and whether it is in the early, mature, or decaying stage of its lifecycle. Additionally, geostationary observations are well-suited for quantifying updraft strength in isolated convective cells, where the time series of minimum cloud-top brightness temperature is expected to be strongly correlated with the intensity of the updraft.

The Doppler capability of EarthCARE's Cloud Profiling Radar (CPR) represents a major innovation, enabling the direct observation of vertical air motion and the terminal fall speeds of hydrometeors. Nonetheless, substantial effort is still required

to fully harness this capability and convert these measurements into actionable items of information for atmospheric science and modeling.

As a next step, a new convection classification framework will be developed using Doppler velocity and radar-derived features. Once established, this classification—when integrated with synergistic geostationary observations—will support the systematic identification of convective regimes and their associated characteristics. This framework will then be applied to generate global-scale statistics.

These efforts will significantly enhance our understanding of convective dynamics at the global scale and are expected to inform and validate high-resolution weather and climate models.

Acknowledgements. The research by AG has been supported by the PANGEA4CalVal project (Grant Agreement 101079201) funded by the European Union. AB has been funded by the Space It Up project funded by the Italian Space Agency, ASI, and the Ministry of University and Research, MUR, under contract n. 2024-5-E.0 - CUP n. I53D24000060005. PK and BPT were supported by the European Space Agency (ESA) under the Clouds, Aerosol, Radiation – Development of INtegrated ALgorithms (CARDINAL) project (RFQ/3-17010/20/NL/AD) and the National Aeronautics and Space Administration (NASA) under the Atmospheric Observing System (AOS) project (Contract number: 80NSSC23M0113).

Author contributions. AG performed the analysis and provided the draft of the manuscript, AB supervised the study and reviewed the paper, PK provided the description of the Doppler velocity analysis and reviewed the paper, BPT provided the updated dataset and reviewed the paper.

Competing interests. At least one of the (co-)authors is a member of the editorial board of Atmospheric Measurement Techniques.

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
