# Peer review of "First insights into deep convection by the Doppler velocity measurements of the EarthCARE Cloud Profiling Radar"

_EGUsphere, 2025_

## Author Comment (AC3)

**ANSWER TO REVIEWER**

**Manuscript egusphere-2025-1914 "First insights into deep convection by the Doppler velocity measurements of the Earth- CARE's Cloud Profiling Radar"**

**Dear Authors,**

**I have taken note of your initial reaction to my review (i.e. suggestions you agree with and those you disagree with). However, I would like to invite you to please consider**

**1) points that are also made by the second reviewer (e.g. regarding the convention for the sign of velocities, which is an easy fix);**

**And**

**2) more importantly, the PDF I attached which details the corrections and edits that I invite you to apply (I only see your reaction to the paragraph posted online, which is only one of many suggested corrections from your title to your references).**

**Respectfully,**

Dear Reviewer,

Thank you very much for your thoughtful feedback, and for the considerable effort you have invested in providing such detailed comments and suggestions.

We truly appreciate the time you took to not only review the manuscript but also to prepare the comprehensive PDF with corrections and edits across the entire document. We acknowledge the importance of the additional points raised—such as the convention for the sign of velocities—and we are currently working on a revised version of the manuscript that carefully implements all the suggested changes, from the title through to the references.

We are committed to addressing your comments in full and will ensure that our revision reflects this.

Sincerely,

Aida Galfione, on behalf of all Authors

---

## Author Response (AR1)

**Response to Associate Editor and Reviewers**

July 21, 2025

Manuscript egusphere-2025-1914 "First insights into deep convection by the Doppler velocity measurements of the Earth-CARE's Cloud Profiling Radar"

**1 Reviewer 1 Comments:**

This article describes an analysis of convective events as observed by the radar of the ESA/JAXA EarthCARE mission. This radar uses a novel technology in space, which is its Doppler capability. The authors do a great job of

- 1. illustrating the quality of the EarthCARE observations and
- 2. tying radar observations to underlying atmospheric dynamics.

The Authors' expertise allows them to juggle between data products and select the most appropriate to analyze various aspects of convection. The task at hand is challenging because vigorous convection often means significant attenuation, large velocities that are aliased, multiple-scattering etc., especially at millimeter wavelengths like the one of EarthCARE's radar.

We thank the reviewer for their careful reading of the manuscript and for the thoughtful and constructive comments, which helped us improve the clarity and focus of the paper.

I suggest major revisions of the article before it can be published. This is motivated by the fact that

- the article is quite long (Section 2 should be subdivided and unnecessary material should be left out).
- there is some work needed in terms of editing to avoid distracting the reader from the main message of your article (such distractions are unfortunate because you obviously did a lot of good work!).

For instance Lines 204-224 are really well written. Could you maintain that standard throughout the article, or have that co-author re-read the article?

We will work on improving Section 2 by splitting it into subsections and removing sentences that are not essential to understanding the topic. In addition, to reduce the overall length of the manuscript, all co-authors will re-read the text to improve its focus and avoid unnecessary digressions.

While I appreciate your use of radar and GEO data, I would recommend also using reanalysis data from ECMWF. In particular, it would be interesting to see if the height of the 0C isotherm is consistent with the melting layer that you observe, and if the ERA5 vertical wind has any updraft.

In convective scenes, due to attenuation and multiple scattering, it is not possible to clearly identify the melting layer. Therefore, the 0°C isotherm from ECMWF was used as a proxy to define the level below which Doppler velocities can reasonably be assumed to be dominated by sedimentation. We believe a comparison with ERA5 data would not be particularly meaningful for the following reasons:

The spatial resolution of reanalysis data is too coarse compared to EarthCARE observations.

- A key strength of the EarthCARE mission is its ability to investigate atmospheric dynamics
  at very high spatial resolution, capturing processes that were previously unresolved by both
  satellite and ground-based instruments.
- In convective systems, updrafts are expected to occur on sub-kilometer scales. While dealiasing Doppler measurements at this resolution is indeed challenging, ERA5 simply cannot resolve such fine-scale dynamics.

Also, I would suggest combining Fig. 7 and 8: they are great but having to flip back and forth between them was yet another distraction from the precious scientific content of your article.

Thank you also for the suggestion to merge figures; unfortunately, Fig. 8 has already three panels and, if merged with Fig. 7, it won't fit in a single page.

Lastly, this is just a convention, but could you please consider showing downward motion as negative velocities? That prevents mental exercises to interpret the Doppler observations.

As is conventional for spaceborne radar measurements, positive Doppler velocity indicates particles falling toward the ground. For this reason, we have chosen to retain the current color scale convention.

All the grammar errors have been corrected in the revised version of the manuscript as suggested in the document attached to the comment.

From the attached document (questions and comments answered already above are omitted):

- Please explain how Doppler radar allows to answer the scientific question of paragraph1, before you describe EarthCARE. Doppler radars, in principle, are able to measure the vertical air motion within clouds, thus giving an insight to understand convective dynamics and convective mass flux. This statement has been added in the abstract of the revised version of the paper.
- While you are reviewing all the spaceborne radar missions, why not mention RAINCUBE's
  Ka-band radar? It did observe quite a lot of convection. The RAINCUBE mission has been
  added to the list of previous radar mission, observing convective events.
- You should consider citing the seminal articles from the European and Japanese teams that describe the Doppler radar of Earthcare (Kumagai et al., Illingworth et al.) References added in the revised version of the paper. Thank you for the suggestion.
- Section 2 is way too long for a reader with limited attention span. Please consider either reducing it, or dividing it into subsections. Section 2 has been divided into three subsections, following the suggestion of the reviewer.
- References are Professor-Kollias-heavy, which is not a slight. However, a lot of work has been done prior to that, going back to Meneghini and Kozu, or Marzoug and Amanyec. Also these references have been added to the revised version of the paper.
- The caption of Fig.1 mentions the Pacific, which is it? Please add the latitude, longitude and times to the X-axis of your figures. That helps figure out over which part of the world the data are from as well as the direction of travel of the aircraft. All Figs. in the manuscript have been updated, as well as their captions, according to this suggestion.
- Why don't you show surface echo and clutter? This is useful info as well. The clutter and surface echo have been added to complete the figures in the updated version of the manuscript.
- This threshold is very high (referring to Doppler velocity showed only where reflectivity; 15dBZ)! Isn't the noise floor in the whereabouts of -35 dbZ? If so, you are really focusing on SNR > 14 dB. Is there a reason for that? No, the noise floor for EarthCARE is -21 dBZ (that defines the reflectivity at SNR=0 dB). In order to produce good Doppler estimates, reflectivities above such thresholds should be used. At negative SNRs the quality of the Doppler measurements very quickly deteriorates.

- (Referring to along-track averaging of Doppler velocity, from 1-km to 4-km.) Please clarify: even though the averaging is done on the complex- valued quantities (lag-1 PP correls) the resulting Doppler velocity may still be aliased. The aliasing is generally reduced because the high velocities encountered at the km-scale are averaged out at 4 km scale. As suggested however, the averaged Doppler velocity is not immune to aliasing.
- The value of  $V_N$  varies along the orbit (because of the PRF). What was the  $V_N$  for these data? (Referring to Fig. 3) The Nyquist velocity for this piece of orbit was 5.08 m/s. It has been added to the text, along with the PRF, for all the analysed scenes.
- Computing the spectral width like this is better than using the spectral width posted in the EC-CPR data because of its prohibitive broadening, right? If so, please clarify.(Referring to Doppler velocity standard deviation, computed with a 3kmx2km running window). No, this is not an estimate of the spectral width (which is typically of the order of 3 m/s due to spectral broadening associated to the platform movement). This is an estimate of the local variability of the Doppler velocity field. High standard deviations are expected to be associated to convective clusters.
- How do you determine the residual error? Do you have a truth/reference? The estimate of the residual errors is based on previous simulation studies based on synthetic scenes where the truth is known from the model output.

**Dear Authors,**

I have taken note of your initial reaction to my review (i.e. suggestions you agree with and those you disagree with). However, I would like to invite you to please consider

- 1. points that are also made by the second reviewer (e.g. regarding the convention for the sign of velocities, which is an easy fix);
- 2. more importantly, the PDF I attached which details the corrections and edits that I invite you to apply (I only see your reaction to the paragraph posted online, which is only one of many suggested corrections from your title to your references).

**Respectfully,**

**Dear Reviewer,**

Thank you very much for your thoughtful feedback, and for the considerable effort you have invested in providing such detailed comments and suggestions. We truly appreciate the time you took to not only review the manuscript but also to prepare the comprehensive PDF with corrections and edits across the entire document. We acknowledge the importance of the additional points raised—especially those that align with the second reviewer's comments, such as the convention for the sign of velocities—and we are currently working on a revised version of the manuscript that carefully implements all the suggested changes. We are committed to addressing your comments in full and will ensure that our revision reflects this.

**2 Reviewer 2 Comments:**

This manuscript describes initial Doppler velocity measurements made by the EarthCARE Cloud Profiling Radar. These initial measurements are exciting and are the initial "pay back" for years of analysis and feasibility studies this team has made exploring the challenges of making Doppler velocity measurements from space. This manuscript is appropriate for Atmospheric Measurement Techniques and will need some minor changes before being ready for publication.

We appreciate the reviewer's careful reading of the manuscript and their constructive comments, which have contributed to improving the clarity and focus of the paper.

**General Comments.**

This reviewer's comments are aimed at clarifying text that could be confusing to the reader. In general, the manuscript is well written. However, there are a couple paragraphs in the middle of the manuscript that are not of the same quality as the rest of the manuscript and will need editing and clarification (details are described below). One concern in these paragraphs is the inclusion

and analysis of Doppler velocity estimates in the regions below multiple scattering, which, I believe, should not have valid atmospheric observations.

**Specific Comments.**

- 1. Line 74. Please correct satellite speed (7.6 km/s). Thank you for highlighting the typo. It has been corrected in the revised version of the manuscript.
- 2. Line 99 and Fig. 1a. Please clarify for the reader whether the spatial resolution of the reflectivity measurements shown in Fig. 1a is at 500-m, 4-km, or some other spatial resolution. In Fig. 1a, the resolution of the reflectivity measurements is 1km. It has been clarified in the text of the revised version of the manuscript.
- 3. Line 102. Please clarify for the reader, is the averaging immune to velocity folding, or is the lag-1 velocity estimator immune to velocity folding? Or, is this statement even necessary? The average is performed in the complex space, to keep it as less sensitive as possible to the velocity folding.
- 4. Line 122. Please inform the reader the value of the Pulse Repetition Frequency and the Nyquist velocity for the examples shown in the manuscript. Frame 1752E (Fig. 3):  $V_N = 5.08$  m/s; PRF=6.38 kHz. Frame 1760E (Fig. 1):  $V_N = 5.09$  m/s; PRF=6.38 kHz. This has been added in the revised version of the manuscript.
- 5. Line 126. Please inform the reader that this estimated maximum value of 6.5 m/s is obtained only when using a radar operating at W-band and that larger reflectivity-weighted mean velocities are measured when using radars operating at lower frequencies. (Readers may be more familiar with Ka-, Ku-, or X-band airborne radars; or Ka-, K-, C-, or S-band ground based radars.) Because Doppler velocities are reflectivity-weighted, and at W-band non-Rayleigh scattering effects tend to reduce the reflectivity of large particles, the maximum reflectivity-weighted terminal velocity at W-band does not exceed 6.5 m/s. We have clarified this point in the revised manuscript.
- 6. Line 136. The phrase 'requires knowledge' is the incorrect phrase to use here because we will never "know" the exact Doppler terminal fall speed (aka, reflectivity-weighted mean fall speed) of the hydrometeors within the radar resolution volume. This sentence is shifting from observations to a retrieval algorithm, so a more appropriate phrase to use here is 'requires parameterization', or some other expression that reflects that the Doppler terminal fall speed is not measured. This is clarified in the revised paper.
- 7. Line 136. Please clarify the text. As written, the phrase "... $V_T^D$  can be..." is equivalent to the phrase "...it could be done, but was not done in this study". This is clarified in the revised version of the manuscript.
- 8. Line 135-137. After reviewing comments 6 and 7, maybe the discussion of retrieving air motion will be confusing to the reader because air motion is not retrieved in this manuscript. Possibly, the sentences from lines 135 to 137 can be deleted. Thank you for spotting this, probably best choice is to delete it.
- 9. Lines 143 to 188. The paragraphs from line 143 through 188 are not of the same quality as other paragraphs in this manuscript. These paragraphs contain grammar errors, errors in logic, and a change in variable notation. These paragraphs need to be rewritten and then proof-read for consistency with the rest of the manuscript. A couple major concerns (and not all concerns) include:
  - Line 170, the text is, "... $V_D$  is positive indicating the presence of an updraft." This is inconsistent with Equation (1) that defines positive values as downward motion. It is an error, the convention is to have positive sign for downward velocity, negative velocities are updrafts. We are correcting it in the revised version of the paper. Equation (1) thus is correct.
  - Figures 4 and 5. The variable  $V_{SED}$  is shown in Fig. 4 and 5, but it is not described in the body. Also, is  $V_{SED}$  the same as  $V_T^D$ ? Yes, they are the same variable. It is clarified in the revised text.

- Line 185, Fig. 4b and 4c below 11 km, and Fig. 5b and 5c below 14 km. Are the authors suggesting that the Doppler velocity measurements below the reflectivity 'knee' corresponding to the height region below multiple scattering are valid and represent atmospheric observations? The authors will need to describe how the change in phase of signals coming from non-radial directions are representative of motions along the radial direction. Doppler velocity measurements in regions affected by multiple scattering cannot be considered reliable. Although these regions were expected to exhibit significantly more noise, this is not always observed. This aspect needs to be further investigated in future studies. By adopting a conservative approach, the Doppler velocity values in such parts of convective cell profiles should not be trusted.
- 10. Lines 257 to 259 and Figure 8. Where are the green bars in Fig. 8a? Added the corrected figure in the revised version of the manuscript.
- 11. Figure 8. What are the symbols in Fig. 8a and 8b? The symbols in Fig. 8 are intended to guide the reader in matching the imagery from different instruments. Indeed, they are reported both on the CPR image and on the MSG image (accounting for parallax correction). A sentence for clarification to this point has been added in the revised version of the manuscript.
- 12. Figure 8. Can you please label Cell 1 and Cell 2 in the figure to help follow the discussion in the text? Done in the revised paper.
- 13. Line 316, the phrase "... unprecedented view of convective motions on a global scale" is incorrect and very misleading. The satellite makes nadir measurements and is in an orbit around the globe. These measurements are not at a "global scale". Also, this work shows images of a few individual precipitation events that do not represent motions on the global scale. Therefore, the phrase can be reduced to, "... unprecedented view of convective motions." We agree, it has been updated in the revised version of the manuscript. In the updated conclusion we will mention the new global perspective.

---

## Author Response (AR2)

**ANSWER TO THE EDITOR**

Manuscript egusphere-2025-1914 "First insights into deep convection by the Doppler velocity measurements of the Earth- CARE's Cloud Profiling Radar"

**Public justification (visible to the public if the article is accepted and published):**

Further minor revisions are required to the manuscript in response to the second set of reviews from the two reviewers.

Additional private note (visible to authors and reviewers only):

Dear authors, thank you for your revisions to your manuscript in response to the reviewers' comments. While both reviewers are positive on the overall merits of your paper, they have requested further changes. Please take a look at the second set of reviews they have provided, including the supplement provided by Reviewer 1, and provide a second revision to the manuscript and a response to their additional comments. In particular:

Reviewer 1 previously provided a detailed annotated version of your manuscript as a supplement to the first review and has expressed some frustration that many of the comments provided were not taken into account. Please take a close look at the new supplement provided by this reviewer and include in your response a point-by-point summary of whether you have taken into account what the reviewer has requested, and if not please provide your argument for not doing so. (I would also recommend you look back at the original supplement, but I am not requesting a point-by-point response to that.) I agree that there were many comments provided, particularly in the supplement accompanying this reviewer's first review, but these comments were all made in good faith to help you improve the manuscript. I agree with the reviewer that the title of the article should be changed (slightly) and that "Doppler" should have a capital "D", and would like to see your response to the other comments in the new annotated manuscript. However, please note that I agree with the authors that the Doppler sign convention matches the EarthCARE convention, and it is OK not to include the height of the melting layer from ERA5.

Reviewer 2 is concerned particularly that you need to convey to the reader more clearly that end-users should not trust velocities in the presence of multiple scattering. Three action items are provided in the new review, and I agree with them all. Comments 1 and 2 apply both to Figure 4 and Figure 5: the captions refer to blue circles when they should be referring to yellow circles. The point where the yellow circles stop indicates where multiple scattering starts to become important, so it would be appropriate to provide (for example) grey shading below this height behind panels 2 and 3 of Figs. 4 and 5 to indicate unreliable Doppler. And finally a comment in the text, probably the conclusion, is needed to state that the Doppler signal is not reliable in the presence of multiple scattering.

**Dear Editor,**

We thank you and the reviewers for the time and effort spent on evaluating our manuscript. We appreciate the constructive feedback and have carefully revised the paper in accordance with the comments provided in the second round of reviews. Below, we provide a detailed point-by-point response to all reviewer comments, indicating the changes made in the revised manuscript.

Sincerely,

The authors

**ANSWER TO REVIEWER 1**

Manuscript egusphere-2025-1914 "First insights into deep convection by the Doppler velocity measurements of the Earth- CARE's Cloud Profiling Radar"

**Dear Authors.**

First, I want to thank you for this revised manuscript.

As stated during the first round, your work presents interesting results about a very interesting mission, and I have no doubt that it will benefit the scientific community.

I am suggesting major revisions of the draft only because many of my suggested corrections from Round 1 were not taken into account (especially in Sections 2.1, 2.2 and 3). I found myself copying them again from Round 1.

This is not to harm your article but quite the opposite as I tried to write as many edits as I could. For instance, during Round 1, I asked to please revise the title of the article, but that was not done. I asked to capitalize the name Doppler in the references but in vain. And, I still do not see the added value of Fig.5, i.e. the unique message that it contains and that is not already conveyed by Fig.4. My comments (many of them copied from R1) are in the document attached and I hope you'll find them useful.

**Regards**

**Dear Reviewer,**

We would like to express our sincere gratitude for your careful evaluation of our revised manuscript and for the constructive and detailed comments you have provided.

We apologize for any oversight in the previous revision that led to some of your earlier suggestions not being fully addressed. We truly appreciate the time and effort you have dedicated to reviewing our work and for reiterating these valuable points.

We have carefully reviewed all of your current comments, including those reiterated from the first round, and we will ensure that the revised version of the manuscript fully incorporates your feedback. We are confident that these improvements will significantly enhance the quality and clarity of our paper.

Thank you once again for your insightful remarks and for contributing to the improvement of our study. With kind regards,

First insights into deep convection by the Doppler velocity measurements of the EarthCARE's Cloud Profiling Radar

or EathCARE's CPR

The title has been changed according to the suggestions.

Title changed to: "First insights into deep convection by the Doppler velocity measurements of the EarthCARE Cloud Profiling Radar"

The Tropical Rainfall Measuring Mission (TRMM), developed by the National Aeronautics and Space Administration (NASA) and the National Space Development Agency of Japan (NASDA), introduced the first spaceborne radar space, a 13.8 GHz Precipitation Radar (PR) (Kummerow et al. 1998; Kummerow et al. 2000). The TRMM PR was operational from

Number: 1 Author: Subject: Cross-Out Date: 8/17/2025 2:17:54 AM you've already written that it's a spaceborne radar

This has been removed in the revised version of the paper.

convective cores and updrafts (Takahashi et al. 2017a). Findings indicate stronger convective cores and lower entrainment rates over land, enabling higher-altitude particle transport. However, CloudSat's narrow across-track sampling (1.4 km cross-track) mits representation of spatially heterogeneous deep convective cores (DCCs). To mitigate biases, CloudSat data have been integrated with passive sensors, such as MODIS cloud top temperature, for improved convective characterization (Luo et al. 2008; Luo et al. 2010; Luo et al. 2014; K. Yang et al. 2023).

Number: 2 Author: Subject: Comment on Text Date: 8/18/2025 11:02:35 PM

Please review the grammar of this sentence (as suggested during round 1 of the review).

Maybe something like: "Because CloudSat was a nadir-looking radar and did not scan across its track, it was limited in its ability to capture the 3-dimensional spatial heterogeneity of DCCs".

Corrected with the suggested sentence.

An example of CPR observations of a deep convective system is shown in Fig. 1. The CPR observations were collected on September 19, 2024, over Vestern Africa on a descending (daytime) orbit. Here, CPR Level 2a (L2a) C-PRO data products

Number: 1 Author: Subject: Highlight Date: 8/18/2025 11:21:14 PM
Thank you for taking into account my comment and adding the latitude.

Would it be possible to have the longitude on top of the second subfigure instead of the latitude repeated?

For instance, to know if we're over Western Africa or the Pacific, one needs the longitude...

The longitude has been added to the Figure. Here below the updated version of Fig. 1.

1 Fig. 1a the 10 dBZ echo top height is very close to the 10 km height for a significant part of the deep precipitating system. In two areas (1710-1730 km and 1890-1910 km along track), the 10 dBZ echo top height is well above the 10 km height and

| Subject: Comment on Text Date: 8/18/2025 | Author: Subject: Comment on Text Date: 8/18/2025 11:30:36 |
|------------------------------------------|-----------------------------------------------------------|
|------------------------------------------|-----------------------------------------------------------|

The comma has been added in the revised version of the manuscript.

110 is 500 m, thus, a total of nine CPR Doppler velocity estimates (their respective real and imaginary parts of the lag-1 pulse pair estimator) have been averaged (Kollias et al. 2023). This averaging is immune to velocity folding. Before the along track averaging, the CPR Doppler velocities have been corrected for antenna mispointing (Puigdomènech Treserras et al. 2025) and non-uniform beam filling (NUBF) Doppler velocity biases (Kollias et al. 2014b; Sy et al. 2014).

| Number: 1        | Author:   | Subject: Highlight     | Date: 8/23/2025 8:29:04 PM |  |  |
|------------------|-----------|------------------------|----------------------------|--|--|
| My question from | Round 1 o | f the review has still | not been addressed.        |  |  |

Please clarify: even though the averaging is done on the complex-valued quantities (lag-1 PP correls) the resulting Doppler velocity may still be aliased.

Does it mean that by averaging over such a long distance that you accept that you beat down the value of the velocity to a level much smaller than the Nyquist velocity? If so is it still useful in the presence of convection/updrafts? Please clarify.

**Added in the revised version of the manuscript:**

"The averaging operation has to be performed in the lag-1 space, in order to avoid the cancellation due to opposite sign in the velocity space, that would lead to a wrong estimation of the Doppler velocity. Averaging over a larger number of pulses reduces aliasing but does not eliminate it, meaning that the Doppler velocity estimates remain susceptible to aliasing errors. Conversely, using a 4-km integration length constrains the ability to resolve the variability within convective cores, which typically occurs at sub-kilometer scales."

The nadir-pointing CPR Doppler velocity  $V_D$  represents the sum of the vertical air motion  $W_{AIR}$  and the reflectivity-

Number: 3 Author: Subject: Comment on Text Date: 8/23/2025 6:16:21 PM

Noted. I have chosen not to fight the battle of the sign convention of teh Doppler velocity.

- 2: Sorry for that, the comments are addressed in the revised version of the manuscript.
- 3: We stick with previous convention as also recommended by the editor: negative velocity upwards (updrafts) and positive velocity downward (downdrafts).

The Value can only take positive values (downward motion) while the Walk Erm can take both positive (downdraft) and

Number: 4 Author: Subject: Highlight Date: 8/23/2025 6:17:04 PM

term V\_T^D

Number: 5 Author: Subject: Highlight Date: 8/23/2025 6:17:09 PM

term WAIR

**Corrected in the revised version of the manuscript.**

negative (updraft) values. The majority of the observed  $V_D$  in Fig. 1b are positive. This implies that the  $V_D^{\text{cl}}$  magnitude is higher than that of the embedded  $W_{AIR}$  updrafts. This suggests the presence of negligible vertical air motions ( $|W_{AIR}|

Ligure 3. (a) CPR reflectivity during a deep convective event stem on September 18, 2024 over Western Africa (Frame 1752E). The blue

Number: 2

Author:

Subject: Cross-Out Date: 8/23/2025 6:40:35 PM

**Corrected in the revised version of the manuscript.**

profiles will be shown in later figures. (b) Be CPR Doppler velocity measurements after a 4-km along-track integration (Kollias et al. 2023).

Author: Subject: Cross-Out Date: 8/23/2025 6:42:09 PM

Number: 3 Author: Subject: Cross-Out Date (it follows a period "." so should be capitalized)

**Corrected in the revised version of the manuscript.**

Number: 4

Author: Subject: Comment on Text

Date: 8/23/2025 6:43:12 PM

Thank you for using the subsections. They are really helpful.

150 The complexity of the  $V_D$  profiles in deep convection is examined using a sample deep convective cloud (DCC) observed by the PR (Fig. 3). The DCC is located between 1265 and 1300 km along track and is characterized by overshooting cloud

Number: 5

Author: Subject: Highlight Date: 8/23/2025 6:44:57 PM

Latitude and longitude axes added to the figure, location (Western Africa, frame 1752E) added in the text in the revised version of the manuscript.

indicate the CPR gates where the Doppler velocity estimates are considered unaffected by multiple scattering. The green triangle indicates the height of the maximum radar reflectivity. (b) The 4-km CPR Doppler velocity profile Green filled circles) and the 1-km CPR Doppler velocity profile (gray filled circles). The black dashed vertical lines indicate the CPR Nyquist Doppler velocity. (c) The unfolded 4-km CPR

Subject: Highlight Date: 8/23/2025 6:50:20 PM Author:

Thank you for taking my comment into account and varying the markers

2 km vertically, centered on each pixel, to capture Doppler velocity variations in both the along-track and across-track by ppler velocity directions.

Number: 2

Author: Subject: Cross-Out Date: 8/23/2025 6:49:23 PM

**Corrected in the revised version of the manuscript.**

170 is presented in Fig. 4a. The blue-filled circles mark the CPR range gates where Doppler velocity estimates are considered unaffected by multiple scattering. Additionally,  $V_D$  estimates near the cloud top are excluded reflectivity falls below -15 dBZ.

Number: 3

Subject: Highlight Date: 8/23/2025 6:53:03 PM

**Corrected in the revised version of the manuscript.**

Beginning with the 4-km profile: near the cloud top,  $\square$  by  $V_D$  is negative, indicating the presence of a weak updraft. Below 14 km, 2 turns positive, which may indicate the presence of large hydrometeors with high sedimentation velocity and/or a downdraft, resulting in an apparent downward motion. The abrupt jump of about 10 m/s in the profile at 12.5 km is attributed

Number: 1 Author: Subject: Cross-Out Date: 8/23/2025 6:56:38 PM either remove or use "the value of V\_D" Subject: Cross-Out Date: 8/23/2025 6:56:14 PM

- 1: Corrected with "the value of V\_D"
- 2: Corrected with "the value of V\_D"
  - a downdraft, resulting in an apparent downward motion. The abrupt jump of about 10 m/s in the profile at 12.5 km is attributed to velocity aliasing. In general if the absolute value of the difference between two consecutive Doppler measurements between the Nyquist velocity, then adding  $\pm 2 V_N$  to one of the velocity produces a smoother 4 rofile. Due to the noisiness of the

Number: 3 Author: Subject: Cross-Out Date: 8/23/2025 6:58:38 PM exceeds: it's the abs value of the diffce

Author: Subject: Highlight Date: 8/23/2025 7:00:47 PM

yes but you also need to know which of the 2 points is the baseline (i.e. not unfolded) and which is not. This is actually a significant part of the logic underlying the dealiasing.

Please clarify in the text.

**3: corrected in the updated version of the manuscript**

4: We agree with reviewer on this. The major challenge in Doppler velocity unfolding arises from the absence of reliable boundary conditions, particularly within convective systems. In stratiform precipitation, it is generally reasonable to assume that motions below the melting layer are predominantly positive (downdrafts), which allows the unfolding procedure to be initiated upward from the top of the bright band. In convective environments, however, such an assumption is not valid. Furthermore, multiple scattering effects often compromise the reliability of measurements in the lower portions of the profiles. As a result, only the upper section of each profile can be considered sufficiently trustworthy. Therefore, the unfolding is performed in a top-down manner, computing velocity differences between successive gates from the highest gate downward, using as a boundary condition the uppermost gate with a sufficiently high signal-to-noise ratio (SNR).

This has been clarified in the text, in the section where the dealiasing is discussed.

| measuremen                    | its, the ident | ification of a fold        | is not straightforward and there will be some ambiguity for successive points in the                                                                                                                                                                                                                                                                                                                                                                                                                                                                                                                                                                                                                                                                                                                                                                                                                                                                                                                                                                                                                                                                                                                                                                                                                                                                                                                                                                                                                                                                                                                                                                                                                                                                                                                                                                                                                                                                                                                                                                                                                                          |
|-------------------------------|----------------|----------------------------|-------------------------------------------------------------------------------------------------------------------------------------------------------------------------------------------------------------------------------------------------------------------------------------------------------------------------------------------------------------------------------------------------------------------------------------------------------------------------------------------------------------------------------------------------------------------------------------------------------------------------------------------------------------------------------------------------------------------------------------------------------------------------------------------------------------------------------------------------------------------------------------------------------------------------------------------------------------------------------------------------------------------------------------------------------------------------------------------------------------------------------------------------------------------------------------------------------------------------------------------------------------------------------------------------------------------------------------------------------------------------------------------------------------------------------------------------------------------------------------------------------------------------------------------------------------------------------------------------------------------------------------------------------------------------------------------------------------------------------------------------------------------------------------------------------------------------------------------------------------------------------------------------------------------------------------------------------------------------------------------------------------------------------------------------------------------------------------------------------------------------------|
| profile with                  | iumps in 6     | close to $7_{\rm M}$ (e.g. | at 4-km integration $5$ ngth values within 1 m/s from $V_N$ are potential foldings). In                                                                                                                                                                                                                                                                                                                                                                                                                                                                                                                                                                                                                                                                                                                                                                                                                                                                                                                                                                                                                                                                                                                                                                                                                                                                                                                                                                                                                                                                                                                                                                                                                                                                                                                                                                                                                                                                                                                                                                                                                                       |
| -                             | јатро т. 🗖     | , crose to A (e.g.         | tale of the first term of the |
| Number: 5                     | Author:        | Subject: Highlight         | Date: 8/23/2025 7:06:06 PM                                                                                                                                                                                                                                                                                                                                                                                                                                                                                                                                                                                                                                                                                                                                                                                                                                                                                                                                                                                                                                                                                                                                                                                                                                                                                                                                                                                                                                                                                                                                                                                                                                                                                                                                                                                                                                                                                                                                                                                                                                                                                                    |
| length,                       |                | , , ,                      |                                                                                                                                                                                                                                                                                                                                                                                                                                                                                                                                                                                                                                                                                                                                                                                                                                                                                                                                                                                                                                                                                                                                                                                                                                                                                                                                                                                                                                                                                                                                                                                                                                                                                                                                                                                                                                                                                                                                                                                                                                                                                                                               |
|                               |                |                            | Date: 8/23/2025 7:00:15 PM                                                                                                                                                                                                                                                                                                                                                                                                                                                                                                                                                                                                                                                                                                                                                                                                                                                                                                                                                                                                                                                                                                                                                                                                                                                                                                                                                                                                                                                                                                                                                                                                                                                                                                                                                                                                                                                                                                                                                                                                                                                                                                    |
| Number: 6                     | Author:        | Subject: Cross-Out         | Date: 6/23/2023 7:00:13 FW                                                                                                                                                                                                                                                                                                                                                                                                                                                                                                                                                                                                                                                                                                                                                                                                                                                                                                                                                                                                                                                                                                                                                                                                                                                                                                                                                                                                                                                                                                                                                                                                                                                                                                                                                                                                                                                                                                                                                                                                                                                                                                    |
| Number: 6 V_D Please be consi |                | ,                          | Date: 0/23/2023 / .00.13 FW                                                                                                                                                                                                                                                                                                                                                                                                                                                                                                                                                                                                                                                                                                                                                                                                                                                                                                                                                                                                                                                                                                                                                                                                                                                                                                                                                                                                                                                                                                                                                                                                                                                                                                                                                                                                                                                                                                                                                                                                                                                                                                   |

**5, 6, 7: Corrected in the revised version of the manuscript.**

Corrected in the revised version of the manuscript.

This profile represents a different portion of the convective event, where the reliable section of the Doppler velocity profile is substantially smaller compared to that shown in Fig. 4. It is included here to illustrate the increased complexity of the unfolding process in convective regions where multiple scattering effects are significant and to highlight the potential ambiguities that arise when differences between consecutive range gates approach the Nyquist velocity (V\_N).

Fig. 5c shows the unfolded 1-km and 4-km  $V_D$  profiles. In  $\bigoplus$  e case, the section of the 4-km averaged  $V_D$  from the cloud top to 195  $\supseteq$  height of 13.8 km is identified as aliased and corrected by subtracting  $2V_N$ . The unfolded 4-km profile displays a smooth vertical structure. A strong updraft is present above 12 km leight, and its magnitude exceeds 10 m/s near the cloud top.

| Number: 1 | Author: | Subject: Cross-Out | Date: 8/23/2025 7:09:00 PM |
|-----------|---------|--------------------|----------------------------|
| this      |         |                    |                            |
| Number: 2 | Author: | Subject: Cross-Out | Date: 8/23/2025 7:09:21 PM |
| a         |         |                    |                            |
| Number: 3 | Author: | Subject: Cross-Out | Date: 8/23/2025 7:12:02 PM |

**1,**

| Analysis of Doppler velocity aliasing  Number: 4 Author: Subject: Highlight Date: 8/23/2025 7:12:40 PM  Thanks for splitting the long initial section into subsections  Based on comprehensive statistics from a large dataset of deep precipitating layers, the standard deviation of the along-track gradient of CPR radar reflectivity is 1.2 dB/km and 2.1 dB/km in stratiform conditions. Figure 6b shows the distribution of the  Number: 5 Author: Subject: Highlight Date: 8/23/2025 7:19:03 PM  As already asked during round 1: I do not understand | 2, 3: Corı | rected i        | n the re    | evised versio          | on of the manuscript.                                                                 |
|---------------------------------------------------------------------------------------------------------------------------------------------------------------------------------------------------------------------------------------------------------------------------------------------------------------------------------------------------------------------------------------------------------------------------------------------------------------------------------------------------------------------------------------------------------------|------------|-----------------|-------------|------------------------|---------------------------------------------------------------------------------------|
| Number: 4 Author: Subject: Highlight Date: 8/23/2025 7:12:40 PM  Thanks for splitting the long initial section into subsections  Based on comprehensive statistics from a large dataset of deep precipitating layers, the standard deviation of the along-track gradient of CPR radar reflectivity is 1.2 dB/km and 2.1 dB/km in stratiform conditions. Figure 6b shows the distribution of the                                                                                                                                                               | a - |                 |             |                        |                                                                                       |
| Thanks for splitting the long initial section into subsections  Based on comprehensive statistics from a large dataset of deep precipitating layers, the standard deviation of the along-track gradient of CPR radar reflectivity is 1.2 dB/km and 2.1 dB/km in stratiform conditions. Figure 6b shows the distribution of the  Number: 5 Author: Subject: Highlight Date: 8/23/2025 7:19:03 PM                                                                                                                                                               | 121.3      | Analysis        | s of Doppl  | er velocity aliasi     | ng                                                                                    |
| Based on comprehensive statistics from a large dataset of deep precipitating layers, the standard deviation of the along-track gradient of CPR radar reflectivity is 1.2 dB/km and 2.1 dB/km in stratiform conditions. Figure 6b shows the distribution of the Number: 5 Author: Subject: Highlight Date: 8/23/2025 7:19:03 PM                                                                                                                                                                                                                                |            |                 |             |                        |                                                                                       |
| gradient of CPR radar reflectivity is 1.2 dB/km and 2.1 dB/km in stratiform conditions. Figure 6b shows the distribution of the Number: 5 Author: Subject: Highlight Date: 8/23/2025 7:19:03 PM                                                                                                                                                                                                                                                                                                                                                               | Than       | nks for splitti | ng the long | initial section into s | ubsections                                                                            |
| gradient of CPR radar reflectivity is 1.2 dB/km and 2.1 dB/km in stratiform conditions. Figure 6b shows the distribution of the Number: 5 Author: Subject: Highlight Date: 8/23/2025 7:19:03 PM                                                                                                                                                                                                                                                                                                                                                               |            |                 |             |                        |                                                                                       |
| gradient of CPR radar reflectivity is 1.2 dB/km and 2.1 dB/km in stratiform conditions. Figure 6b shows the distribution of the Number: 5 Author: Subject: Highlight Date: 8/23/2025 7:19:03 PM                                                                                                                                                                                                                                                                                                                                                               |            |                 |             |                        |                                                                                       |
| Number: 5 Author: Subject: Highlight Date: 8/23/2025 7:19:03 PM                                                                                                                                                                                                                                                                                                                                                                                                                                                                                               | Bas        | sed on con      | nprehensiv  | e statistics from a    | large dataset of deep precipitating layers, the standard deviation of the along-track |
|                                                                                                                                                                                                                                                                                                                                                                                                                                                                                                                                                               | 205 gra    | idient of CI    | PR radar re | flectivity is 51.2 d   | B/km and 2.1 dB/km in stratiform conditions. Figure 6b shows the distribution of the  |
|                                                                                                                                                                                                                                                                                                                                                                                                                                                                                                                                                               |            |                 |             |                        |                                                                                       |
| As already asked during round 1: I do not understand                                                                                                                                                                                                                                                                                                                                                                                                                                                                                                          |            |                 |             |                        |                                                                                       |
| what are these two numbers. Are they limit values? modal values? guantile values? How to interpret them?                                                                                                                                                                                                                                                                                                                                                                                                                                                      |            |                 |             |                        |                                                                                       |

These values refer to the standard deviation of the dataset containing the along-track reflectivity gradient (in dB/km). The values in the new version of the manuscript have been recomputed and are now right (standard deviation of the reflectivity gradient is 5.25 dB/km and 1.67 dB/km in convective and in stratiform conditions, respectively). This gradient is computed as the difference in reflectivity between successive along-track gates, using a horizontal resolution of 1 km. High variability in the CPR reflectivity gradient—and consequently a high standard deviation—is expected in convective regions.

Number: 7 Author: Subject: Comment on Text Date: 8/23/2025 7:21:51 PM Sorry to insist on this, but I just don't understand what is the reference that you use to estimate these errors? IS it that you expect the signals to be 0 and measure a difference? Please clarify

This point has been clarified in the revised version of the paper: the Doppler velocity error is estimated by multiplying the reflectivity gradient by a correction factor. For EarthCARE, this factor is 0.165 m/s /(dB/km), resulting in a corresponding velocity bias. The residual random error is 20% of the velocity bias, as from literature. These values have been corrected accordingly to the comment 6 above.

230 finite horizontal scales. Moreover, the presence of multiple scattering and strong attenuation further limits the applicability of simplifying assumptions such as  $W_{AIR} \approx 2$  below the melting layer  $\frac{1}{1}$  eight.

| Number: 1 | Author: | Subject: Cross-Out Date: 8/ | /23/2025 7:23:38 PM        |
|-----------|---------|-----------------------------|----------------------------|
| Number: 2 | Author: | Subject: Comment on Text    | Date: 8/23/2025 7:23:36 PM |

**1, 2: corrected in the revised version of the manuscript.**

Geostationary satellites today provide a quasi-global coverage in a dide, common set of wavelengths bross-different agencies, ranging from visible shortwaves to infrared (IR) (Fiolleau et al. 2024). Over the past decade, the capabilities of geostationary

Number: 3

Author: Subject: Cross-Out Date: 8/23/2025 7:29:32 PM

**3, 4: corrected in the revised version of the manuscript**

ranging from visible shortwaves to infrared (IR) (Fiolleau et al. 2024). Over the past decade, the capabilities of geostationary satellites have increased significantly in terms of spectral diversity to observations. A synergistic effort is currently underway to

| Number: 5 | Author: Subject: Cross-Out | Date: 8/23/2025 7:29:52 PM

**Corrected in the revised version of the manuscript.**

**From line 234 to 240, the text has been changed to:**

"Over the past decade, the capabilities of geostationary satellites have increased significantly in terms of spectral diversity and enhanced spatial and temporal resolution of observations. The synergistic use of these measurements with range resolved cloud and precipitation profiles offers a unique opportunity to the scientific and meteorological community. In particular, compared to CloudSat, EarthCARE offers finer horizontal resolution and is equipped with a Doppler radar. This enables, for the first time, global observations of in-cloud vertical velocities, thus deeper insights into convective storm lifecycle processes and the corresponding environmental responses."

the Atlas Mountains in North Africa, during a descending (daytime) orbit (Fig. 7). Figure 7a how the visible band (0.6 µm) radiance from the MSG satellite, with the EarthCARE satellite ground track overlaid in red. Figure 7b displays the MSI brightness temperatures in the clean infrared band (10.8 µm). Several convective cloud complexes are detected, and some deep

Number: 9 Author: Subject: Comment on Text Date: 8/23/2025 7:32:30 PM shows (already from round1)

**Corrected in the revised version of the manuscript.**

Figure 7. (a) Radiance from channel 1 (0.6 µm) of MSG, on November 7th, 2024 at 13:45 UTC. The EarthCARE ground track, corrected for the parallax is shown by the red line. (b) he MSI IR channel data from the EarthCARE satellite. The overpass time is 13:43 UTC on November 7th, 2024, frame 2530D. The red line is the satellite ground track. The segment between the two stars is plotted in Fig. 8.

| Number: 1   | Author: | Subject: Cross-Out | Date: 8/23/2025 7:34:20 PM |
|-------------|---------|--------------------|----------------------------|
| T Number: 2 | Author: | Subject: Cross-Out | Date: 8/23/2025 7:34:18 PM |
| as          |         |                    |                            |
| Number: 3   | Author: | Subject: Cross-Out | Date: 8/23/2025 7:34:35 PM |

**1, 2, 3: Corrected in the revised version of the manuscript.**

developed anvil reaches an altitude of 10 km and is primarily detrained southward from the main convective core. Between 48.4° and 38.5° latitude, a stratiform region is identified, characterized by a continuous reflectivity echo extending from the

Number: 4 Author: Subject: Cross-Out Date: 8/23/2025 7:38:18 PM latitudes of 38.4 and 38.5 (Round 1)

**Corrected in the revised version of the manuscript.**

surface to cloud-top heights above 10 km. The melting layer is clearly marked by the presence of a bright band. North of this stratiform region, around 8.7° latitude, a stronger convective core is observed, featuring a thick reflectivity column keeedling 15 dBZ. This core reaches nearly 12 km in echo-top height, overshooting the anvil cloud top. Furthermore, the cluster of

Number: 5 Author: Subject: Cross-Out Date: 8/23/2025 7:39:27 PM
that exceeds
(you already use "featuring" in your sentence)

**Corrected in the revised version of the manuscript.**

Number: 6 Author: Subject: Underline Date: 8/23/2025 7:37:23 PM

**Corrected in the revised version of the manuscript.**

Between 36.5° and 36.75° Tittude, high cloud-top echoes reaching 8–10 km, reflectivities exceeding 10 dBZ, and significant attenuation collectively indicate a well-developed convective system. Further south, between 36° and 36.3° tittude, the lower height of the reflectivity echo, combined with the presence of very high reflectivity between 4 km and 6 km, suggests a con-

| Number: 7   | Author: | Subject: Highlight | Date: 8/23/2025 8:00:51 PM |
|-------------|---------|--------------------|----------------------------|
| of latitude |         |                    |                            |
| Number: 8   | Author: | Subject: Highlight | Date: 8/23/2025 7:57:25 PM |
| of latitude |         |                    |                            |

vective cell still in ledevelopmental phase—likely in an earlier stage of its let eyele. In Fig. 8c, the orange line represents the number of Doppler velocity foldings per profile, while the blue line indicates the number of pixels per profile that exceed the stratiform range threshold ([-2 3] m/s).

| Number: 1 | Author: | Subject: Cross-Out | Date: 8/23/2025 7:58:25 PM |
|-----------|---------|--------------------|----------------------------|
|           |         |                    |                            |
| Number: 2 | Author: | Subject: Cross-Out | Date: 8/23/2025 7:58:46 PM |
| lifecycle |         |                    |                            |

**Corrected in the revised version of the manuscript.**

Using the CloudSat methodology, a DCC would have been identified in Cell 1 between 38.5° and 38.7° [3] titude (green bar on the right in Fig. 8a) whereas only the central tower in Cell 2, located around 36.5° latitude, would be classified as deep

| Number: 3   | Author: | Subject: Highlight | Date: 8/23/2025 8:01:47 PM |
|-------------|---------|--------------------|----------------------------|
| of latitude |         |                    |                            |

**Corrected in the revised version of the manuscript.**

To place the CPR convective cloud snapshots within the context of their die cycle, observations from EUMETSAT's MSG satellite are analyzed. Sign-9 and 10 display the corresponding MSG SEVIRI 1.5 rapid scan frames from channel 9 (10.8 µm)

**Corrected in the revised version of the manuscript.**

In contrast, Cell 2 is more isolated, which facilitates more effective tracking and blows its evolution to be observed from the early stages. Fig. 10 reveals a secondary cold spot on the southwest side of the cell, which later merges with the main

| Number: 6  | Author: | Subject: Highlight | Date: 8/23/2025 8:10:19 PM |
|------------|---------|--------------------|----------------------------|
| allows for |         |                    |                            |

**Corrected in the revised version of the manuscript.**

Figure 9. Successive images depicting the time evolution of the IR brightness temperature field 10 minutes before (a), 5 minutes before (b), closest in time (c), 5 minutes after (d) and 10 after (e) Earth CARE overpass 2530D on November 7th, 2024, zoom on cell 1. The colors represent the brightness temperature from channel 9 (10.8 µm) MSG rapid scans. Black solid line present the ground track of Earth CARE, corrected for parallax (dashed line is the original ground track). The black markers correspond to Fig. 8a. The black star is the minimum of the brightness temperature that is tracked.

| Number: 1               | Author: | Subject: Highlight | Date: 8/23/2025 8:14:22 PM |
|-------------------------|---------|--------------------|----------------------------|
| measured by             |         |                    |                            |
| ,                       |         |                    |                            |
|                         |         |                    |                            |
| Number: 2               | Author: | Subject: Highlight | Date: 8/23/2025 8:15:00 PM |
| Number: 2
represents | Author: | Subject: Highlight | Date: 8/23/2025 8:15:00 PM |

300 based on cloud-top cooling rates. While tobac tracking provides valuable temporal context, it annot capture the fine-scale vertical variability and internal dynamics revealed by the EC-CPR.

Number: 3 cannot

Author: Subject: Highlight Date: 8/23/2025 8:12:39 PM

Corrected in the revised version of the manuscript.

This discussion reinforces the limitations of relying solely on geostationary infrared cooling rates the characterizing convection. While IR observations are effective at capturing relatively large and isolated updrafts near cloud tops, embedded

Number: 4 to characterize (Round 1)

Author: Subject: Cross-Out Date: 8/23/2025 8:13:22 PM

Corrected in the revised version of the manuscript.

as the horizontal and vertical extent, of convective updrafts and downdrafts. Doppler velocity-based detection of convective cores is compared with traditional reflectivity-based methods. This comparison is expected to inform a revision of the detection 320 criteria used in previous spaceborne radar 1 udies.

Number: 1 Subject: Highlight Date: 8/23/2025 8:20:12 PM Author: Please state briefly how that will be the case based on the findings of your article?

Lines 318-320 text has been changed to:

"Our case studies give evidence of differences in the detection of convective cores when Doppler velocity based instead of reflectivity based criteria are used. Future studies should investigate the impact of new Doppler velocity based criteria onto the climatology of occurrences of convective cores across Earth."

Kollias, P., S. Tanelli, A. Battaglia, and A. Tatarevic (2014b). "Evaluation of EarthCARE cloud profiling radar bppler velocity measurements in particle sedimentation regimes". In: Journal of Atmospheric and Oceanic Technology 31.2. DOI: 10.1175/JTECH-D-11-00202.1 (cit. on pp. 3, 5).

Number: 1 Subject: Underline Date: 8/17/2025 2:29:00 AM Author:

Corrected in the revised version of the manuscript.

Sy, O. O., S. Tanelli, N. Takahashi, Y. Ohno, H. Horie, and P. Kollias (2014). "Simulation of EarthCARE spaceborne by Products and P. Kollias (2014)." 570 using ground-based and airborne data: Effects of aliasing and nonuniform beam-filling". In: IEEE Transactions on Geoscience and Remote Sensing 52.2. DOI: 10.1109/TGRS.2013.2251639 (cit. on pp. 3, 5).

Number: 1 Author: Subject: Underline Date: 8/17/2025 2:35:02 AM Doppler (it's a Last name, like Galfione)

**ANSWER TO REVIEWER 2**

Manuscript egusphere-2025-1914 "First insights into deep convection by the Doppler velocity measurements of the Earth- CARE's Cloud Profiling Radar"

This revised manuscript is much improved over the previous revision.

I believe the section describing Doppler velocities during multiple scattering still needs some improvement. The current version still contains errors, and the manuscript should clearly state that Doppler velocity estimates are not reliable at range gates with multiple scattering. As written, the manuscript states that Doppler velocities in range gates containing multiple scattering are valid and should be trusted. This conclusion contrasts with the author's response to my comment #9.

I believe that stating the Doppler velocities are not reliable during multiple scattering is a positive finding and end-users should be warned to not trust velocities during multiple scattering.

We thank the reviewer for this important clarification and for emphasizing the need to better communicate the impact of multiple scattering on Doppler velocity reliability. We agree that Doppler velocity estimates in range gates affected by multiple scattering should not be considered reliable.

**Action Items:**

1. The figure caption for Figure 5 does not match Figure 5. The yellow circles in panel 5a are labeled "Reliable Doppler", but the text says, 'The blue filled circles... are considered unaffected by multiple scattering." Thus, the figure caption says that the blue reflectivity pixels below 12 km are not affected by multiple scattering, yet the yellow circles above 13 km are labeled as reliable. These two statements cannot be correct at the same time. Please fix the figure caption and/or the figure.

Solved. I changed the Fig. but I didn't update carefully the caption.

2. Figure 5b and 5c need some kind of marking to indicate where the radial velocities are not reliable. I suggest shading the height-velocity domain where the radial velocities are considered not reliable.

The shading to highlight the effect of Multiple scattering has been added to Fig. 5 and Fig. 4.

3. I believe that some of the text the authors made in the reply to reviewer should be included in the manuscript to indicate that the authors know that the Doppler velocities are not reliable during multiple scattering. This text was: "Doppler velocity measurements in regions affected by multiple scattering cannot be considered reliable. Although these regions were expected to exhibit significantly more noise, this is not always observed. This aspect needs to be further investigated in future studies. By adopting a conservative approach, the Doppler velocity values in such parts of convective cell profiles should not be trusted."

This text has been added in the paragraph where the multiple scattering effect is discussed:

"Doppler velocity measurements within regions affected by multiple scattering cannot be considered reliable. Although a marked reduction in the correlation between successive pulses—and

consequently an increase in Doppler velocity noise—is expected in these regions, such behavior is not always observed. This inconsistency warrants further investigation in future studies. Adopting a conservative approach, Doppler velocity values within these portions of convective cell profiles should therefore be treated with caution or excluded from quantitative analysis."